## [Decision Letter · Decision Letter 0]

11 Jan 2023

PONE-D-22-31323Inhibition of the PCR by Genomic DNAPLOS ONE

Dear Dr. Morley,

Thank you for submitting your manuscript to PLOS ONE. After careful consideration, we feel that it has merit but does not fully meet PLOS ONE’s publication criteria as it currently stands. Therefore, we invite you to submit a revised version of the manuscript that addresses the points raised during the review process.

We look forward to receiving your revised manuscript.

Kind regards,

Ruslan Kalendar

Academic Editor

PLOS ONE

Journal Requirements:

4. We note that you have a patent relating to material pertinent to this article. Please provide an amended statement of Competing Interests to declare this patent (with details including name and number), along with any other relevant declarations relating to employment, consultancy, patents, products in development or modified products etc. 

Please confirm that this does not alter your adherence to all PLOS ONE policies on sharing data and materials, as detailed online in our guide for authors http://journals.plos.org/plosone/s/competing-interests by including the following statement: ""This does not alter our adherence to  PLOS ONE policies on sharing data and materials.” 

If there are restrictions on sharing of data and/or materials, please state these. Please note that we cannot proceed with consideration of your article until this information has been declared.

7. In your Data Availability statement, you have not specified where the minimal data set underlying the results described in your manuscript can be found. PLOS defines a study's minimal data set as the underlying data used to reach the conclusions drawn in the manuscript and any additional data required to replicate the reported study findings in their entirety. All PLOS journals require that the minimal data set be made fully available. For more information about our data policy, please see http://journals.plos.org/plosone/s/data-availability.

Reviewers' comments:

Reviewer's Responses to Questions

**Comments to the Author**

1. Is the manuscript technically sound, and do the data support the conclusions?

Reviewer #1: Partly

Reviewer #2: Partly

2. Has the statistical analysis been performed appropriately and rigorously? 

Reviewer #1: I Don't Know

Reviewer #2: Yes

3. Have the authors made all data underlying the findings in their manuscript fully available?

Reviewer #1: No

Reviewer #2: Yes

4. Is the manuscript presented in an intelligible fashion and written in standard English?

Reviewer #1: Yes

Reviewer #2: Yes

5. Review Comments to the Author

Reviewer #1: 

I have reviewed the manuscript by Latham et al entitled "Inhibition of the PCR by Genomic DNA".

This manuscript shows that Genomic DNA inhibits PCR. While the experimental results appear to support the conclusions, the manuscript has major flaws that require extensive revision prior to publication, as described below.

First, the manuscript was submitted to PLOS ONE, an interdisciplinary journal; PLOS ONE has a very broad readership, the majority of whom are not medical professionals. However, the manuscript is written as if it were submitted to a journal of diagnostic medicine, which makes it very difficult for non-medical experts, including myself, to read it. It is necessary to add enough explanation to be understood by researchers in a wide range of fields. In this sense, Methods in particular needs a complete revision. For example, it was not possible for me to find the sequence of primers used in the PCR. This makes it impossible to determine the validity of the paper.

Second, the terminology needs to be revised. In many parts of the manuscript, quantitative real-time PCR is referred to simply as PCR or conventional PCR without explanation, which hinders understanding.For most readers, simply mentioning PCR or conventional PCR refers to so-called first-generation PCR and never to realtime PCR.

For the above reasons, this manuscript needs extensive revision before publication.

Reviewer #2: 

Comments:

This article discusses the factors that cause nonspecific amplicons during conventional PCR due to primer interaction with non-target genomic DNA. The discussed factors include primer design, annealing temperature, G/C content and polymerase concentrations. The authors extracted target DNA from clinical blood samples and compared the PCR amplification performance of commonly used primers versus patient-specific primers. According to their finds, patient-specific primers can reduce non-specific amplification and decrease the MRD detection sensitivity down to 10-4. Also, the amplification performance can be improved via increasing polymerase concentrations, a higher annealing temperature between primer and target DNA, and a lower G/C content. However, this work contains no significant innovation, and the written needs to be further improved. A major revision is suggested.

1. Can the author provide metrics when stating MRD detection levels? The authors stated “10-4” “10-6” in the manuscript but did not include metrics.

2. Please consider including the full name of “HAT-PCR”. Does it mean “high-annealing-temperature PCR”?

3. The author used the term PCR in the manuscript, but I believe they meant qPCR. It is recommended to clarify or correct it.

4. Some agreements are stated without reference, such as “It has a limit of detection of 10-6 when 20 μg of DNA are assayed. The sensitivity of conventional PCR, performed according to Euro MRD guidelines and using a patient-specific forward primer and a recommended reverse J primer, was studied following development of HAT-PCR. An individual primer pair usually detected levels of MRD down to 10-4 but frequently failed to detect lower levels”.

5. The genomic DNA is first described as non-target genomic DNA in the Results section. It is suggested to clarify what genomic DNA is in the abstract and introduction.

6. It is recommended to re-draw Figure 1-B panel. When template DNA is in the range of 0.2-2 ng, the dotted lines are difficult to distinguish.

7. Figure 2 is not understandable. Please consider adding a legend. The samples with added patient primers can be categorized into three groups. Can the authors provide the reason? What is different between the 6 patient samples?

8. Can the authors provide more information about genomic DNA and DNA extracted from clinical blood samples? Are they both the same length? Will the result change if different pieces of genomic DNA are added in the PCR reactions?

9. Although the manuscript claims that patient-specific primers can result in lower detection limits, designing patient-specific primers in real-world diagnostic settings is impractical.

6. PLOS authors have the option to publish the peer review history of their article (what does this mean?). If published, this will include your full peer review and any attached files.

Reviewer #1: No

Reviewer #2: No

---

## [Author Response · Author response to Decision Letter 0]

15 Feb 2023

1. Please ensure that your manuscript meets PLOS ONE's style requirements….

This has been done to the best of my abilities

2. Please provide additional details regarding participant consent…… 

Ethics now in Methods, consent was written, data anonymised.

Corrected

4. We note that you have a patent relating to material pertinent to this article…… 

Done.

“This patent does not alter our adherence to PLOS ONE policies on sharing data and materials.” 

This information is included in the cover letter. I note that you will change the online submission form on our behalf.

5. Your ethics statement should only appear in the Methods section of your manuscript….. 

Done

6. Please include captions for your Supporting Information files at the end of your manuscript, and update any in-text citations to match accordingly……

Done. Please check that has been done correctly

7. In your Data Availability statement, you have not specified where the minimal data set underlying the results described in your manuscript can be found….. 

The minimal data set is in the manuscript and the Supporting Information.

Comments to the Author

Reviewer #1

First, the manuscript was submitted to PLOS ONE, an interdisciplinary journal; PLOS ONE has a very broad readership, the majority of whom are not medical professionals. However, the manuscript is written as if it were submitted to a journal of diagnostic medicine, which makes it very difficult for non-medical experts, including myself, to read it. It is necessary to add enough explanation to be understood by researchers in a wide range of fields. In this sense, Methods in particular needs a complete revision. For example, it was not possible for me to find the sequence of primers used in the PCR. This makes it impossible to determine the validity of the paper.

Our paper is directed not only to medical professional but also to anyone who uses PCR. This would encompass a fairly wide scientific readership. Some degree of scientific depth and complexity is necessary when addressing this readership and this will obviously make it difficult for the reader outside these areas.

The only point which the scientist with biochemical training might have difficulty in grasping would be the calculation of the inhibition index. We have tried to explain this in the Methods and have given a concrete example of the calculation in the Supplementary Data.

With regard to the issue of primer sequences, the sequences of the EuroMRD generic reverse primers are documented in refs 2 and 3 and those of the HAT-PCR reverse primers are documented in Table 1 of ref 5. Forward primers are patient-specific so the sequence of each is unique. However, more relevant than the sequences is the issue of primer design. For EuroMRD primers, this is given in refs 2 and 3 and is considered in the Discussion in lines 262-272. For HAT-PCR primer design is given in ref 5 and is now given explicitly in the Discussion in lines 275-278. It is then followed in lines 278-281 by the specific details of the design features as performed in the present study. 

Second, the terminology needs to be revised. In many parts of the manuscript, quantitative real-time PCR is referred to simply as PCR or conventional PCR without explanation, which hinders understanding.For most readers, simply mentioning PCR or conventional PCR refers to so-called first-generation PCR and never to realtime PCR.

We point out that “quantify” or “quantification” is referred to in the first line of the Abstract, the first line of the Introduction and at numerous other points, and that any PCR directed to produce a numerical MRD value must be quantitative. 

However, in order to make “PCR” even more explicit we have now changed PCR to qPCR at many points in the Introduction, Methods and Results, and have rearranged the Methods so that the first paragraph of the techniques states that all PCRs were quantitative PCRs. However we have retained “PCR” in the Discussion as we are often considering the PCR in general.

Reviewer #2: 

1. Can the author provide metrics when stating MRD detection levels? The authors stated “10-4” “10-6” in the manuscript but did not include metrics.

An MRD level is the proportion of the cell type of interest in the total population of cells being analysed and it is a pure number without a unit.

2. Please consider including the full name of “HAT-PCR”. Does it mean “high-annealing-temperature PCR”?

This has been done in line 54

3. The author used the term PCR in the manuscript, but I believe they meant qPCR. It is recommended to clarify or correct it.

Please see the response to Reviewer 1

4. Some agreements are stated without reference, such as “It has a limit of detection of 10-6 when 20 μg of DNA are assayed. The sensitivity of conventional PCR, performed according to Euro MRD guidelines and using a patient-specific forward primer and a recommended reverse J primer, was studied following development of HAT-PCR. An individual primer pair usually detected levels of MRD down to 10-4 but frequently failed to detect lower levels”.

The references are now given.

5. The genomic DNA is first described as non-target genomic DNA in the Results section. It is suggested to clarify what genomic DNA is in the abstract and introduction.

The term “genomic DNA” is widely used and refers to cellular DNA in general. It can be isolated by nonspecific means. In terms of the PCR the target is the sequence which is that part of genomic DNA to which to which the primers are designed to amplify. The remaining genomic DNA is non-target DNA. Primers may also bind to regions of non-target genomic DNA in a nonspecific fashion and to a variable degree.

6. It is recommended to re-draw Figure 1-B panel. When template DNA is in the range of 0.2-2 ng, the dotted lines are difficult to distinguish.

The point of the figure is to illustrate severe (1A) and mild (1B) inhibition. With mild inhibition and a large amount of DNA the lines by their nature will overlap and become difficult to distinguish, as the PCR requires few cycles to reach threshold and the magnitude of nonspecific amplification is not sufficient to produce inhibition. Only when the amount of DNA is low and the PCR requires many cycles to reach threshold does the inhibitory effect become sufficiently great to become manifest as a separation of the lines. This contrasts with the situation in Fig 1A where the effect of added genomic DNA is manifest at relatively high levels of DNA.

7. Figure 2 is not understandable. Please consider adding a legend. The samples with added patient primers can be categorized into three groups. Can the authors provide the reason? What is different between the 6 patient samples?

The legend has been modified to try and clarify. The point to be illustrated is the inhibitory effect that was seen with all 6 pairs of patient primers. The sequence of the forward primer differs from patient to patient and the reverse primer may also differ. Variation in primers causes variation in nonspecific binding to genomic DNA and this leads to variation in the degree of inhibition. To clarify we now state in line 51 that each forward primer is unique. 

8. Can the authors provide more information about genomic DNA and DNA extracted from clinical blood samples? Are they both the same length? Will the result change if different pieces of genomic DNA are added in the PCR reactions?

Genomic DNA is the generic term for the DNA in cells that contains the genetic information. It represents the DNA in the nucleus and mitochondrion. In broad terms it is the same in all somatic cells and tissues but minor differences in cells and tissues can arise due to somatic mutation and disease. Blood and marrow samples are relatively rich in lymphocytes in which the various IG and TCR genes undergo rearrangements in the process of developing the immune repertoire. These rearrangements provide the targets for MRD quantification in disorders of neoplastic lymphocytes.

Total genomic DNA is added to the PCR reaction. The primers bind to the target, more or less specifically.

9. Although the manuscript claims that patient-specific primers can result in lower detection limits, designing patient-specific primers in real-world diagnostic settings is impractical.

 Patient -specific primers have been used for real-world quantification for over 3 decades, particularly in Europe and by the EuroMRD consortium. The technique is not easy but it can be done. It is not the use of patient-specific primers result in lower detection limits. It is the optimal design of the patient-specific primers and the optimal design of the PCR that enable lower detection.

1. Please ensure that your manuscript meets PLOS ONE's style requirements….

This has been done to the best of my abilities

2. Please provide additional details regarding participant consent…… 

Ethics now in Methods, consent was written, data anonymised.

Corrected

4. We note that you have a patent relating to material pertinent to this article…… 

Done.

“This patent does not alter our adherence to PLOS ONE policies on sharing data and materials.” 

This information is included in the cover letter. I note that you will change the online submission form on our behalf.

5. Your ethics statement should only appear in the Methods section of your manuscript….. 

Done

6. Please include captions for your Supporting Information files at the end of your manuscript, and update any in-text citations to match accordingly……

Done. Please check that has been done correctly

7. In your Data Availability statement, you have not specified where the minimal data set underlying the results described in your manuscript can be found….. 

The minimal data set is in the manuscript and the Supporting Information.

Comments to the Author

Reviewer #1

First, the manuscript was submitted to PLOS ONE, an interdisciplinary journal; PLOS ONE has a very broad readership, the majority of whom are not medical professionals. However, the manuscript is written as if it were submitted to a journal of diagnostic medicine, which makes it very difficult for non-medical experts, including myself, to read it. It is necessary to add enough explanation to be understood by researchers in a wide range of fields. In this sense, Methods in particular needs a complete revision. For example, it was not possible for me to find the sequence of primers used in the PCR. This makes it impossible to determine the validity of the paper.

Our paper is directed not only to medical professional but also to anyone who uses PCR. This would encompass a fairly wide scientific readership. Some degree of scientific depth and complexity is necessary when addressing this readership and this will obviously make it difficult for the reader outside these areas.

The only point which the scientist with biochemical training might have difficulty in grasping would be the calculation of the inhibition index. We have tried to explain this in the Methods and have given a concrete example of the calculation in the Supplementary Data.

With regard to the issue of primer sequences, the sequences of the EuroMRD generic reverse primers are documented in refs 2 and 3 and those of the HAT-PCR reverse primers are documented in Table 1 of ref 5. Forward primers are patient-specific so the sequence of each is unique. However, more relevant than the sequences is the issue of primer design. For EuroMRD primers, this is given in refs 2 and 3 and is considered in the Discussion in lines 262-272. For HAT-PCR primer design is given in ref 5 and is now given explicitly in the Discussion in lines 275-278. It is then followed in lines 278-281 by the specific details of the design features as performed in the present study. 

Second, the terminology needs to be revised. In many parts of the manuscript, quantitative real-time PCR is referred to simply as PCR or conventional PCR without explanation, which hinders understanding.For most readers, simply mentioning PCR or conventional PCR refers to so-called first-generation PCR and never to realtime PCR.

We point out that “quantify” or “quantification” is referred to in the first line of the Abstract, the first line of the Introduction and at numerous other points, and that any PCR directed to produce a numerical MRD value must be quantitative. 

However, in order to make “PCR” even more explicit we have now changed PCR to qPCR at many points in the Introduction, Methods and Results, and have rearranged the Methods so that the first paragraph of the techniques states that all PCRs were quantitative PCRs. However we have retained “PCR” in the Discussion as we are often considering the PCR in general.

Reviewer #2: 

1. Can the author provide metrics when stating MRD detection levels? The authors stated “10-4” “10-6” in the manuscript but did not include metrics.

An MRD level is the proportion of the cell type of interest in the total population of cells being analysed and it is a pure number without a unit.

2. Please consider including the full name of “HAT-PCR”. Does it mean “high-annealing-temperature PCR”?

This has been done in line 54

3. The author used the term PCR in the manuscript, but I believe they meant qPCR. It is recommended to clarify or correct it.

Please see the response to Reviewer 1

4. Some agreements are stated without reference, such as “It has a limit of detection of 10-6 when 20 μg of DNA are assayed. The sensitivity of conventional PCR, performed according to Euro MRD guidelines and using a patient-specific forward primer and a recommended reverse J primer, was studied following development of HAT-PCR. An individual primer pair usually detected levels of MRD down to 10-4 but frequently failed to detect lower levels”.

The references are now given.

5. The genomic DNA is first described as non-target genomic DNA in the Results section. It is suggested to clarify what genomic DNA is in the abstract and introduction.

The term “genomic DNA” is widely used and refers to cellular DNA in general. It can be isolated by nonspecific means. In terms of the PCR the target is the sequence which is that part of genomic DNA to which to which the primers are designed to amplify. The remaining genomic DNA is non-target DNA. Primers may also bind to regions of non-target genomic DNA in a nonspecific fashion and to a variable degree.

6. It is recommended to re-draw Figure 1-B panel. When template DNA is in the range of 0.2-2 ng, the dotted lines are difficult to distinguish.

The point of the figure is to illustrate severe (1A) and mild (1B) inhibition. With mild inhibition and a large amount of DNA the lines by their nature will overlap and become difficult to distinguish, as the PCR requires few cycles to reach threshold and the magnitude of nonspecific amplification is not sufficient to produce inhibition. Only when the amount of DNA is low and the PCR requires many cycles to reach threshold does the inhibitory effect become sufficiently great to become manifest as a separation of the lines. This contrasts with the situation in Fig 1A where the effect of added genomic DNA is manifest at relatively high levels of DNA.

7. Figure 2 is not understandable. Please consider adding a legend. The samples with added patient primers can be categorized into three groups. Can the authors provide the reason? What is different between the 6 patient samples?

The legend has been modified to try and clarify. The point to be illustrated is the inhibitory effect that was seen with all 6 pairs of patient primers. The sequence of the forward primer differs from patient to patient and the reverse primer may also differ. Variation in primers causes variation in nonspecific binding to genomic DNA and this leads to variation in the degree of inhibition. To clarify we now state in line 51 that each forward primer is unique. 

8. Can the authors provide more information about genomic DNA and DNA extracted from clinical blood samples? Are they both the same length? Will the result change if different pieces of genomic DNA are added in the PCR reactions?

Genomic DNA is the generic term for the DNA in cells that contains the genetic information. It represents the DNA in the nucleus and mitochondrion. In broad terms it is the same in all somatic cells and tissues but minor differences in cells and tissues can arise due to somatic mutation and disease. Blood and marrow samples are relatively rich in lymphocytes in which the various IG and TCR genes undergo rearrangements in the process of developing the immune repertoire. These rearrangements provide the targets for MRD quantification in disorders of neoplastic lymphocytes.

Total genomic DNA is added to the PCR reaction. The primers bind to the target, more or less specifically.

9. Although the manuscript claims that patient-specific primers can result in lower detection limits, designing patient-specific primers in real-world diagnostic settings is impractical.

 Patient -specific primers have been used for real-world quantification for over 3 decades, particularly in Europe and by the EuroMRD consortium. The technique is not easy but it can be done. It is not the use of patient-specific primers result in lower detection limits. It is the optimal design of the patient-specific primers and the optimal design of the PCR that enable lower detection.

---

## [Decision Letter · Decision Letter 1]

3 Apr 2023

Inhibition of the PCR by Genomic DNA

PONE-D-22-31323R1

Dear Dr. Morley,

We’re pleased to inform you that your manuscript has been judged scientifically suitable for publication and will be formally accepted for publication once it meets all outstanding technical requirements.

Kind regards,

Ruslan Kalendar

Academic Editor

PLOS ONE

---

## [Editor Report · Acceptance letter]

13 Apr 2023

PONE-D-22-31323R1 

Inhibition of the PCR by Genomic DNA 

Dear Dr. Morley:

I'm pleased to inform you that your manuscript has been deemed suitable for publication in PLOS ONE. Congratulations! Your manuscript is now with our production department. 

Kind regards, 

on behalf of

Professor Ruslan Kalendar 

Academic Editor

PLOS ONE